# Determinants of Continuance Intention to Use Hearing Aids among Older Adults in Tehran (Iran)

**DOI:** 10.3390/healthcare12040487

**Published:** 2024-02-17

**Authors:** Abdolhakim Jorbonyan, Yadollah Abolfathi Momtaz, Mahshid Foroughan, Saeideh Mehrkian

**Affiliations:** 1Department of Geriatric Health, School of Health, Mazandaran University of Medical Sciences, Sari 48157-33971, Iran; a.jourbonian@mazums.ac.ir; 2Iranian Research Center on Aging, The University of Social Welfare and Rehabilitation Sciences, Tehran 19857-13871, Iran; ma.foroughan@uswr.ac.ir; 3Malaysian Research Institute on Ageing, University Putra Malaysia, Serdang 43400, Selangor, Malaysia; 4Department of Audiology, The University of Social Welfare and Rehabilitation Sciences, Tehran 19857-13871, Iran; sa.mehrkian@uswr.ac.ir

**Keywords:** post-acceptance model (PAM), hearing aids, continuance intention

## Abstract

The present study seeks to evaluate the factors determining the continuance intention to use hearing aids in older adults. This cross-sectional study was carried out in 2021. The technology post-acceptance model (PAM) framework was used to develop a model for the continuance intention to use hearing aids. In total, 300 hearing aid users aged ≥60 years, who were selected via a randomized stratified sampling method, completed the self-evaluation tools used in this study. With a mean age of 71.38 years (SD = 8), the participants comprised 50.7% and 49.3% females and males, respectively. The path analysis results showed that the continuance intention to use hearing aids was positively and significantly influenced by the actual use of hearing aids, the perceived benefits, satisfaction, confirmation, self-efficacy in using hearing aids, an extraverted personality trait, self-perceived hearing handicap, and perceived social support. The main results of the present study can help hearing care providers develop a better understanding of older users to design effective rehabilitation strategies and ensure their continuance intention to use hearing aids.

## 1. Introduction

Presbycusis or age-related hearing loss is the most common hearing loss in humans, and is observed in around 10.9–17.6% of people aged 60–69 years, 41.9–51.2% of those aged 80–89 years, and 52.9–64.9% people aged ≥90 years [1,2]. Experts suggest that no compensation for or correction of hearing loss can impact various mental and emotional health aspects of older adults. For instance, prolonged deprivation of the sense of hearing brings about changes in the central nervous system that eventually lead to dementia [3,4].

Using hearing aids is the most convenient solution to compensate for hearing loss in older adults. Hearing aids can largely make up for the communicational problems and defects stemming from hearing loss and increase the health-related quality of life in the elderly [5]. Yet, many older adults are not willing to receive hearing aids and use hearing rehabilitation services [6]. Furthermore, the results of several studies suggest that around 30–50% of the elderly do not use them regularly or set them aside permanently [7,8,9]. Thus, identifying and understanding the role of the real determinants of the actual and continuous use of hearing aids can significantly help us develop effective interventions and promote the use of hearing aids in the elderly suffering from hearing loss [10].

Three review studies have summarized the results of other relevant studies and reported that factors such as socioeconomic status, educational level, the severity of hearing loss, insertion gains, perceived hearing handicap, the personality traits of the elderly, and positive social support could play vital roles in the acceptance and use of hearing aids [11,12,13]. These studies highlighted the role of psychosocial factors, attitudes, and beliefs, and suggested that the reasons behind not using hearing aids were beyond the features and performance of hearing aids, given the advancements in the technology and appearance of modern hearing aids [14]. Hence, audiologists have partnered with health psychologists in recent years to explain the predictors of a person using hearing aids and other hearing rehabilitation interventions through the application of social psychology and health models and theories [15].

A review of the studies that have so far used these models—e.g., the health belief model [15,16], theory of planned behavior [17], and the theory of reasoned action [18]—indicates that they mostly focus on the factors and behaviors that determine the acceptance intention or the early stages of receiving hearing aids. Some researchers argue that the psychosocial and behavioral aspects influencing the use of technology or the adopting of a new health behavior change over time and across various stages of acceptance [19]. Thus, a model investigating the behaviors affecting the post-acceptance stage of receiving hearing aids—that is, the continuous and actual use of hearing aids—in older adults is among the research gaps that have yet to be addressed, to the best of our knowledge. Therefore, the present study proposes a model to examine the behaviors of hearing aid users in the stages following the prescription of hearing aids.

## 2. Theoretical Background and Research Framework

The determinants of technology acceptance are often investigated in four stages, including (1) familiarization with the technology, (2) the intention to use the technology, (3) the acceptance of the technology, and (4) the post-acceptance stage of the technology [20]. The theories of reasoned action, planned behavior, technology acceptance, and expectation-confirmation are among the most renowned theories examining user behaviors at various stages of technology acceptance [21]. For instance, Cobelli et al. [18] used Ajzen’s theory of reasoned action to identify the factors affecting the intention to use hearing aids. Meister et al. [17] used the planned behavior theory to evaluate the determinants of hearing aid acceptance intention. Laplante et al. [22] used the theory of change stages to present a model for the stage of change in hearing rehabilitation in older adults suffering from hearing loss. Saunders et al. [16] investigated hearing health behaviors based on the framework of the health belief model.

Few studies have relied on these theories to examine the actual use of hearing aids at the post-fitting stage [23,24]. According to researchers, the impact of the perceptions and attitudes of individuals on their use of hearing aids might vary before and after their initial acceptance of hearing aids, due to their experience of using hearing aids and their practical application in daily life [16,25]. In other words, identifying practical and effective methods to improve the use of hearing aids or any other assistive devices would be extremely difficult without an adequate understanding of users’ attitudes and behaviors throughout the whole rehabilitation process.

### 2.1. Expectation-Confirmation Theory (ECT)

Several studies beyond the scope of health have sought to employ technology post-acceptance models to explain users’ behaviors in the post-acceptance stage [26,27,28]. In this group of studies, the ECT (Oliver 1980) is widely used and explains user satisfaction and their intentions toward the continued use of goods and services [29]. This theory suggests that satisfaction with a product is a function of their expectation, performance, and confirmation. This model consists of four interconnected sections; the first stage is the individual’s expectation. At this stage, users form their initial expectations based on their previous experiences, the knowledge available to them, and their interaction with various people regarding specific goods or services, without purchasing or receiving them [30]. The second stage is the perceived benefits of the product’s performance, in which people receive a product if they deem it to be useful [26]. The third stage is the confirmation or rejection of their beliefs. After a period of using the product, users form their perception of the product’s performance based on their initial expectations and decide the extent to which their expectations are met. Confirmation occurs if the product is better than expected, while rejection occurs when the users’ assessments are lower than expected [31]. The fourth stage is satisfaction with the product, in which users’ confirmation of the product informs their sense of satisfaction. Satisfied users generally continue their use of the product, and unsatisfied users might cease their use of the product [32].

Overall, confirmation and satisfaction are the two fundamental principles of the expectation-confirmation theory that influence the intention to repurchase or continue the use of a product [33]. Wong et al. (2004) [34] and Meyer et al. (2014) [35] examined the relationship between the expectation of and satisfaction with hearing aids using the expectation-confirmation model. Wong et al. [34] investigated 42 hearing aid users with an average age of 64.2 years (SD = 14.8) in Hong Kong and reported a positive and significant relationship between confirmation and satisfaction with hearing aids, but they observed no direct relationship between expectations and a satisfaction with hearing aids. Meyer et al. [35] studied 123 hearing aid users with a mean age of 72 years (SD = 7) and found that satisfaction with the hearing aids was associated with confirmation, hearing ability in various environments, appearance features, and ease of use, based on their logistic regression results.

### 2.2. Post-Acceptance Model (PAM)

Given that some constructs of the expectation-confirmation model have been associated with the pre-acceptance stage of the technology, Bhattacherjee (2001) made modifications to this theory and integrated it with Davis’s technology acceptance model (TAM) to propose a PAM to explain the behaviors and beliefs of users at the stage of using the technology [26,36]. This model focuses on the key role of the two concepts of satisfaction and confirmation and their influence on improving the continuance intention to use the technology. In this model, technology confirmation obtained through comparing users’ expectations before using the technology to its real performance increases their satisfaction with technology and leads to their continuance intention of its use. The PAM’s reasoning indicates that the two constructs of satisfaction and confirmation cover the factors influencing the pre-acceptance stage; additionally, the expectation of individuals may vary throughout their use of the technology. Therefore, the PAM interprets and evaluates the expectations at the stage of using the technology as the perceived benefits [26,37]. As a result, the model assumes that the confirmation of technology impacts its perceived benefits and satisfaction. The users then decide whether or not to continue using the technology based on its benefits and their satisfaction with it. Several studies have developed a PAM to investigate factors affecting the continuance intention to use various types of technology [38,39,40].

For instance, Cho et al. [41] employed a PAM to investigate factors influencing the continued use of health applications in 343 participants in South Korea. Their results suggested that 58.6% of the variation in continuance intention to use health applications was explained by the constructs of the model proposed in the study.

### 2.3. Extended Post-Acceptance Model (PAM)

The determinants of technology use may vary depending on the type of the technology, its application, and the user community. Many researchers have recommended the development of technology acceptance models by adding other influential constructs [42]. The results of previous studies show that factors such as self-efficacy in using hearing aids, perceived hearing handicap, personality traits, and perceived social support play determining roles in hearing aid outcomes (use, benefits, and satisfaction) [11,12,13]. Therefore, the present study added these mentioned constructs to the original PAM and developed a model of the continuance intention to use hearing aids based on the PAM’s framework. The following subsections explain the causal relationships between the constructs of the proposed model.

### 2.4. Confirmation

The post-acceptance theory suggests that satisfaction with the technology will increase if its performance is confirmed through its usage [43]. Moreover, confirmation of the technology positively affects its perceived benefits, which means that the user will find the technology useful if its performance is compliant with their expectations [26]. Although the influence of confirmation on the satisfaction with and benefits of hearing aids has yet to be investigated using the PAM, several studies have confirmed this relationship using the expectation-confirmation model [44,45,46].

### 2.5. Perceived Benefits

According to Davis and Venkatesh, perceived benefits refer to the extent to which the individual believes that the use of a system will increase their performance [36,47]. The original technology acceptance model (TAM) emphasizes the direct influence of perceived benefits on a person’s attitude toward a type of technology, which will eventually determine their intention to continue using the technology [48,49]. Since satisfaction is a positive stage of feelings and attitude, the perceived benefits can have a positive relationship with the satisfaction with technology [38]. A significant impact of perceived benefits on satisfaction with hearing aids (0.5–0.83; *p* ≤ 0.05), examined using self-evaluation tools, has been reported in previous studies [50,51]. Moreover, several studies have reported that perceived benefits had a significant relationship with the actual use of hearing aids [7,23,52].

### 2.6. Satisfaction

Satisfaction can be defined as a pleasant feeling and experience when evaluating the product’s performance compared to expectations about it [43]. Satisfaction is an extremely essential factor in the process of hearing aid prescription [53]. Patients who are satisfied with their hearing aids generally use them regularly, whereas a low level of satisfaction is among the main reasons to abandon the use of hearing aids in most patients who do not use their hearing aids [34,54].

### 2.7. Hearing Aid Self-Efficacy

Self-sufficient people generally tend to resist their encountered obstacles and adopt new behaviors to manage their health conditions. Hearing aid self-efficacy can be defined as the level of self-confidence and capability of an individual in using hearing aids [55]. Several studies have demonstrated that the elderly people suffering from hearing impairment who were adequately self-sufficient used hearing aids more successfully. Thus, self-efficacy can be considered among the predictors affecting the benefits, satisfaction, and use of hearing aids [23,46].

### 2.8. Extraverted Personality Trait

Several studies suggest that extroverts benefit more from their hearing aids and have a higher level of satisfaction and hours of hearing aid usage than introverts [56,57,58].

### 2.9. Perceived Social Support

Recent studies have reported considerable correlations between positive social support and longer use of hearing aids. These studies suggest that the accessibility of social support, encouragement, and the presence of associates in rehabilitation programs are factors that eventually increase the successful use of hearing aids in those suffering from hearing loss [53,59].

### 2.10. Self-Perceived Hearing Handicap

According to the World Health Organization, self-perceived hearing handicap refers to the non-hearing consequences, such as emotional distress and restricted social participation, directly caused by a disorder in the hearing structure [60]. The results of several studies indicate that people who more often reported their hearing problems used their hearing aids for longer hours [5,61,62].

### 2.11. Actual Use and Continuance Intention to Use

There is no agreed-upon definition for the actual use of hearing aids in the literature on hearing. However, the definition of actual and successful hearing aid use suggested by Hickson et al. (2014)—reporting at least one hour of daily hearing aid use and receiving the least moderate benefits in situations where the individuals need to hear better—appears to be more rational than the definitions relying merely on the number of hours hearing aids are used for [23]. The continuance intention to use a type of technology refers to the intentions of users to continue their use of a specific technology [26]. This means that the continuance intention to use could predict the future use of technology. At the stage following the acceptance of hearing aids, users use their hearing aids and may either continue or discontinue using them. Thus, the actual use of hearing aids can influence the continuance intention to use them.

## 3. Materials and Methods

### 3.1. Participants

In the present cross-sectional study, the participants were 300 hearing aid users (60 years or older) residing in Tehran. In the sampling process, Tehran was divided into four zones, namely low, lower–moderate, upper–moderate, and high in terms of socioeconomic development levels [63]. Several audiology clinics were then randomly contacted in each zone to collect the address and contact information of the older adults who had received hearing aids from these clinics. Next, the elderly were classified into four categories based on their place of residence. Eventually, 75 people from each category were randomly selected (through a draw) as the research sample group. The inclusion criteria were its having been at least 6 months after their hearing aid prescription (unilateral and bilateral hearing aid fitting), their ability to communicate and good command of the Persian language to respond to the questions, and consent to participate in the study voluntarily. The exclusion criteria were reluctance to keep participating in the study, incomplete completion of questionnaires, and physical or mental discomfort in completing the questionnaires.

### 3.2. Measurement Scales

In the present study, the data were collected using several self-evaluation tools, demographic information forms, and the last recorded audiograms of the individuals. Table 1 explains the details and reliability of the tools used in the present study. Translation and psychometrics of the three questionnaires were carried out before executing the study, as detailed below.

#### 3.2.1. Continuance Intention and Confirmation Scales

Given the lack of a special tool to assess the confirmation and continuance intention to use hearing aids, the present study uses the constructs of the general PAM questionnaire (Bhattacherjee 2001), which is a 12-item tool scored on a five-point Likert scale. This tool has been used in several studies to evaluate the factors affecting the use of technologies such as health applications, information technologies, cellphones, etc. [26,27,37]. Thus, the constructs of continuance intention to use (three items) and confirmation (three items) in this questionnaire were translated into Persian, and their face and content validity were evaluated and confirmed by a group of experts consisting of 10 audiology and geriatric specialists. The Content Validity Ratio (CVR) and Content Validity Index (CVI) of the two scales of confirmation and continuance intention to use were higher than 0.85 and 0.90, respectively, indicating the favorable content validity of the scales. The internal consistency of the questionnaire items was calculated at α = 0.74 and α = 0.83 for continuance intention to use and confirmation, respectively, using Cronbach’s alpha, suggesting the appropriate internal reliability of the two tools.

#### 3.2.2. Hearing Aid Self-Efficacy Questionnaire

Self-efficacy was evaluated using the Measure of Audiologic Rehabilitation Self-Efficacy for Hearing Aids (MARS-HA) [64], consisting of 24 questions and four subscales. To score the questions, the respondents must assign to each question a score on a 10-unit interval scale (0, 10, …, 100%), specifying their level of confidence in their ability. A score of zero means that the respondents feel they have no skill or ability, and a score of 100 means that the respondents are completely confident when performing this skill. Higher scores in this questionnaire indicate higher self-efficacy. This questionnaire was also translated into Persian, and its face and content validity were evaluated and confirmed by a group of experts. The calculated CVR and CVI indices were, respectively, >0.80 and 0.91 for this questionnaire. The internal consistency of the questionnaire items was calculated as α = 0.94 using Cronbach’s alpha.
healthcare-12-00487-t001_Table 1Table 1The characteristics of the questionnaires.ConstructInstrumentsDescription/ScaleCronbach’s αSatisfactionPersian version of Satisfaction with Amplification in Daily Life (SADL) [65].A 15-item questionnaire that evaluates the satisfaction that people feel with their current hearing aids. It quantifies satisfaction using a global score and four subscales: Positive Effect, Service and Cost, Negative Features, and Personal Image. The SADL response scale: 7-point Likert (not at all; a little; somewhat; medium; considerably; greatly; tremendously)0.91Perceived benefitPersian version of Abbreviated Profile of Hearing Aid Performance (APHAP) [66].A 24-item self-assessment inventory that is divided into four subscales that assess communication, including situations in favorable environments (EC scale) and experiences in the presence of noise (BN scale), reverberating rooms (RV scale), and loud sounds (AV scale). The APHAP response scale: A. Always (99%), B. Almost Always (87%), C. Generally, (75%), D. Half the time (50%), E. Occasionally (25%), F. Seldom (12%), G. Never (1%)0.92Actual usePersian version of the International Outcome Inventory for Hearing Aids (IOI-HA) [67].In this study, we defined the actual use of hearing aids based on each participant’s self-report responses to question 1 (average hours of hearing aid use per day) and question 2 (the benefit from hearing aids in the situations where the individual most wanted to hear better) on the IOI-HA. Response scale: 5-point Likert; question 1 (none = 1 … to more than 8 h a day = 5), question 2 (helped not at all = 1 … helped very much = 5)0.85Self-perceived hearing handicapPersian version of the Hearing Handicap Inventory Screening Version for the Elderly (HHIE-S) [68].A 10-item questionnaire that evaluates how an individual perceives the social and emotional effects of hearing impairment. Response scale: 3-point Likert (No = 0; Sometimes = 2; Yes = 4)0.85Perceived social supportThe Iranian version of the Multidimensional Scale of Perceived Social Support (MSPSS) [69].A 12-item measure of the perceived adequacy of social support from three sources: family, friends, and significant otherResponse scale: 5-point Likert (strongly disagree = 0, strongly agree = 5)0.93Extraverted personality traitPersian version of the Ten-Item Personality Inventory (TIPI) [70].In this study, we used questions 1 and 6 of the Ten-Item Personality Inventory that evaluates the personality trait of extraversion.Response scale: a 7-point scale ranging from 1 = strongly disagree to 7 = strongly agree0.64


### 3.3. Data Analysis

Data were analyzed using descriptive statistics and analytical tests, including Pearson’s correlation coefficient, an independent *t*-test, a one-way analysis of variance (ANOVA), and multiple linear regression, using the SPSS V.24 software. The relationships between the main research variables were simultaneously analyzed using SEM (structural equation modeling) (path analysis through maximum likelihood estimation) in AMOS V.24 software. The model fit and the consistency between experimental data and the conceptual model were examined using the goodness of fit (GOF) indices and criteria. In the present study, the GOF indices included X2/Df, RMSEA, GFI, and TLI, which were used to measure the final GOF of the model.

## 4. Results

The mean age of the participants was 71.38 years (SD = 8), ranging between 60 and 95 years. In terms of gender, 49.30% and 50.70% of the respondents were male and female, respectively. Table 2 details the demographic information of the participants. The results of the bivariate analyses indicated that factors such as age (r = −0.16, *p* ˂ 0.001), level of education (F (4, 295) = 12.29, *p* ≤ 0.001), employment status (F (3, 296) = 4.3, *p* ˂ 0.001), and the type of hearing aids (F (2, 297) = 7.09, *p* ≤ 0.001) had significant relationships with the actual use of hearing aids. However, no significant relationship was observed between actual hearing aid use and other audiological factors (*p* > 0.05) (Table 3). Among all the audiological factors and demographic variables (Table 3), the results suggested that only the type of hearing aids had a significant relationship with the continued use of hearing aids (F (2, 297) = 6.87, *p* ≤ 0.001). Descriptive results from the main research variables indicated that 63.3% of hearing aid users had actual use of, while 36.7% failed in the actual use of, their hearing aids. The overall average values of actual hearing aid use and continuance intention to use hearing aids were, respectively, 6.12 (SD = 2.19) and 10.14 (SD = 2.91), which indicate a moderate level of actual use and continuance intention to use hearing aids among the participants, considering the mean values of the questionnaires. The analysis of Pearson’s correlation revealed a significant and positive relationship between the actual use and the continuance intention to use hearing aids (r = 0.67, *p* < 0.01). Table 4 lists the descriptive data and correlation coefficients between the research variables.

### 4.1. Findings from Multiple Linear Regression

The prediction power of all the significant variables in Table 3 and Table 4 (*p*-value < 0.2) in predicting the actual use and continuance intention to use hearing aids was examined using a multivariate analysis. In this study, the assumptions of the linear regression included data normality (all the variables had kurtosis and skewness ranges of −2, 2 according to the tests) and a residual normality test, and the mean standardized residual error was calculated as close to zero for all models, and their standard deviation was calculated as close to one. Durbin–Watson test values confirmed the independence of the errors, meaning the obtained values were between 1.5 and 2.5. The multiple collinearity test revealed that the VIF values for the independent research variables were <3 and the tolerance values were >0.25, indicating no collinearity between the independent research variables. Dummy variables were used to include the qualitative variables in the regression analysis.

Table 5 shows the characteristics and statistics related to the fitting of the regression models. The F value and the significance level of the regression models indicated a statistically significant relationship between the dependent variables—namely actual use and continue intention to use hearing aids—and the independent research variables (*p* < 0.001). In the actual hearing aid use model, the variables included in the equation explained 69% (R^2^ = 69) of the real variations in the actual use of hearing aids by the participants.

The Beta values indicated that, among the demographic variables, high school education level and higher had a positive impact on the actual use of hearing aids (β = 0.08, *p* ≤ 0.05), while the use of analog hearing aids (β = −0.08, *p* ≤ 0.05) and unemployment (β = −0.12, *p* ≤ 0.005) had negative impacts on the actual use of hearing aids. Of the main variables, factors such as perceived benefits (β = 0.22, *p* ˂ 0.001), confirmation (β = 0.26, *p* ˂ 0.001), self-perceived hearing handicap (β = 0.19, *p* ˂ 0.001), and hearing aid satisfaction (β = 0.18, *p* ˂ 0.01) were the best predictors of the use of hearing aids. In the model of continuance intention to use hearing aids, the independent variables explained 60% (R^2^ = 60) of the variation in the continuance intention to use hearing aids. β values in the model indicated that the influence of a low education level (β = −0.10, *p* ˂ 0.04) and the use of analog hearing aids (β = −0.08, *p* ˂ 0.03) had negative impacts on the continuance intention to use hearing aids. Confirmation (β = 0.22, *p* ˂ 0.001) and the actual use of hearing aids (β = 0.43, *p* ˂ 0.01) had the greatest influence on the continuance intention to use hearing aids.

### 4.2. Results from SEM

The values of the GOF indices, after making the necessary modifications based on the modified indices in the final model, resulting from the SEM were CMIN = 2.52, RMSEA = 0.07, GFI = 0.87, and CFI = 0.92, which are suitable GOF indices and reveal the confirmation of the proposed research model. Moreover, Table 6 summarizes the final model’s GOF indices and their recommended values. Standard regression weights, standard error, significance levels, and the critical ratio (CR) were investigated to confirm or reject the causal relationships between the research variables (Table 7). Figure 1 demonstrates the final research model in its standard state.

## 5. Discussion

The present research mainly sought to evaluate the determinants of the continuance intention to use hearing aids in older adults. The mean continuance intention to use hearing aids was at a moderate level, and 36.7% of the participants did not actually use their hearing aids. Using similar criteria, the Hickson et al. (2014) study found that 46% of their participants were unsuccessful in using their hearing aids [23]. Overall, this result from the present study is consistent with some other studies, suggesting no regular use of hearing aids in 30–50% of the elderly [71,72].

### 5.1. Demographic and Audiological Characteristics

The relationships between the actual use and continuance intention to use hearing aids and several demographic variables such as age, gender, and marital status were not significant according to the results of our multivariate analysis. Similarly, some other studies suggest that factors such as age, gender, and marital status cannot determine the amount of use of hearing aids when several other predictors are incorporated into the multivariate model [52,62,73]. However, some studies (e.g., Klyn et al., 2019) reported that women used hearing aids significantly less than men [74], while Ji-Su (2015) found that older adults in the age group of 60–64 used hearing aids for longer times than older groups [75].

Among the demographic variables, a higher education level and job status (pensioner compared to unemployed) significantly predicted the actual and continued use of hearing aids. It is not surprising that those with higher educational levels and socioeconomic status used hearing aids more often due to their greater feeling of their need for hearing aids, as they are able to afford hearing aids, and have a more prominent presence in the community. Likewise, higher education levels and socioeconomic status positively impacted the use of hearing aids in other studies [76,77,78].

In the present study, the actual use of and continuance intention to use hearing aids had no significant relationships with audiological factors such as the severity of hearing loss, a history of using hearing aids, the types of fittings, and the appearance of hearing aids. In terms of the type of hearing aids, however, analog hearing aids negatively influenced their actual use. Confirming this result, three studies reported that the risk of irregular hearing aid use was higher in analog hearing aids than in more advanced signal-processing hearing aids [79,80,81]. Contrary to the results of the present study, several studies argued that higher degrees of hearing loss were associated with a more frequent use of hearing aids [61,76,82]. The present results corroborate those of Ferguson et al. (2016) and Maeda et al. (2016), who reported that the degree of hearing loss did not determine the regular use of hearing aids [46,62]. Given the factors investigated in the present study, it can generally be understood that demographic features and audiological factors cannot be strong predictors for the actual and continuous use of hearing aids in older adults. Although there is a need to examine other audiological factors, the important role of the beliefs, hearing-associated quality of life, and personality traits of the elderly in their actual and continuous use of hearing aids cannot be overlooked in such studies.

### 5.2. Extended PAM-Related Constructs

The present study explained the influence of the main research variables on the continuance intention to use hearing aids through the development of a PAM. The results of the path analysis confirmed the positive impact of the perceived benefits of and satisfaction with hearing aids. This means that hearing aid users perceive higher benefits and satisfaction with their hearing aids if they confirm that their experience of using the hearing aids is compliant with or beyond their initial expectations, which will eventually result in the continuous use of hearing aids [45]. Similarly, a direct relationship between the benefits of and satisfaction with hearing aids has been reported in previous studies [32,46,83]. However, one study found that confirmation was not a strong predictor of hearing aid outcomes [84].

The results of several studies have suggested that most successful hearing aid users had a high self-efficacy in using them, and self-efficacy was a significant factor in the adherence to hearing aids [85,86,87]. The present study found that self-efficacy had the greatest impact on the level of satisfaction, followed by the perceived benefits of hearing aids; however, self-efficacy’s direct relationship with the actual use of hearing aids was not significant. This result is consistent with that of Ferguson et al. (2016), who suggested that self-efficacy was associated with satisfaction with hearing aids but that it had no significant relationship with the frequency of the use of hearing aids [46]. Thus, it can be justified that self-efficacy in using hearing aids increases the level of satisfaction and the perceived benefits of hearing aids in users, which will eventually lead to higher actual and continuous hearing aid use. The present study calculated a mean self-efficacy score of 64.58%, which indicates the moderate self-efficacy of the older adults participating in the study. It is thus recommended that audiology experts enhance the self-efficacy beliefs of the elderly in their hearing rehabilitation sessions through an awareness of the potential factors, such as cognitive disorders, physical abilities, or educational levels, influencing self-efficacy in the older population. The results of several studies suggest that reporting a self-perceived hearing handicap—or in other words, reporting the emotional and social effects of a hearing impairment—in daily life can play a determining role in all aspects of hearing aid use [61,62,88]. In the present study, the path coefficient also indicated that higher scores in the hearing handicap variable were among the determinants of actual hearing aid use, meaning that the individual needs to achieve a significant understanding of the emotional and social impacts of hearing loss to make actual use of their hearing aids. However, several studies suggest that a perceived hearing handicap may be influenced by non-audiological factors such as ethnicity, gender, language, educational level, and the individual’s health conditions. For instance, those of a slightly older age may consider their hearing loss a natural consequence of age and report it less frequently [10,89,90].

There is ample evidence suggesting that specific aspects of the moods and personalities of individuals influence their ability to tackle stress and adapt to new environments [91,92]. The present study found significant and positive relationships between an extraverted personality trait and the perceived benefits, satisfaction, and actual use of hearing aids, as reported in previous studies [58,83,93]. Extroverts gain more benefits from and satisfaction with their hearing aids over time and become more willing to use them continuously due to their active presence in social environments and their use of hearing aids in various environments [94].

Studies that focused on the role of social support in managing hearing loss revealed that older adults with hearing loss who have access to social support networks adapt to their hearing loss faster than others, and use their hearing aids more regularly [13,53,95]. The present study also found social support to be among the determinants of the actual use of hearing aids, suggesting that more support received by individuals from their families, friends, and other significant people in their lives will make them more open to the actual use and continuance intention to use their hearing aids. Several recommendations can be given concerning the role of social support in the actual use of hearing aids. Firstly, encouragement and motivation from others can be a reinforcing factor for the continued use of hearing aids. Secondly, participation in the handling and maintaining of hearing aids (repairing, changing batteries, and helping in taking, wearing, and cleaning hearing aids) by others can be another influential factor encouraging the continued use of hearing aids. Thirdly, social support in the form of emotional support can help hearing aid users adapt to their hearing aids [96].

The results of the path analysis indicated that the perceived benefits of hearing aids could predict a person’s satisfaction with their actual use of hearing aids. To interpret the predictive power of the benefits, if hearing aids act in such a way as to improve the hearing difficulties of users in various hearing conditions, their satisfaction with hearing aids will increase accordingly. High levels of satisfaction with hearing aids will eventually lead to the actual use of hearing aids. However, older adults may remain dissatisfied with their hearing aids for a number of reasons, but their perception of the beneficial performance of hearing aids can steer them toward an actual use and continuance intention to use hearing aids. This finding is confirmed by other studies, in which the perceived benefits and satisfaction in the post-fitting stage of hearing aids were two significant and determining factors in hearing aid use [46,97,98]. Contrary to the benefits that are merely associated with the performance of hearing aids, satisfaction encompasses the appearance, social, psychological, and financial aspects of receiving and using hearing aids [34,99]. The present study also revealed that satisfaction was among the factors determining the rate of the actual use of hearing aids, such that an increased level of satisfaction will increase actual hearing aid usage, which corresponds to reports indicating that satisfaction with hearing aids had a positive relationship with the number of hours of hearing aid use [54,100,101]. However, Aurelio et al. (2012) found no significant relationship between satisfaction and the number of hours of hearing aid use, but they noted that significant results would be obtained in the case of larger sample size [102].

Finally, the results of the present study suggest that the actual use of hearing aids is a strong predictor of the continuance intention to use hearing aids. This means that the successful use of hearing aids by the elderly will increase their intention to use them continuously in the future. Confirming this result, Lee et al. (2020) investigated the key factors influencing the continuance intention to use wearable healthcare devices. Their results revealed that internal factors (knowledge, attitudes, and beliefs) and external factors (technological and social factors) positively impacted actual user behavior, and the continuance intention to use wearable healthcare devices could be improved through actual use behavior [103].

## 6. Limitations and Further Research

Similar to other studies, the present study faced several limitations. First and foremost, the cross-sectional nature of this study makes it difficult to make conclusions about the causalities. Thus, it would be better to collect data longitudinally for a closer investigation of the perceptual factors affecting the actual and continuous use of hearing aids. Secondly, the present study used questionnaire tools for data collection, hence some participants might have refrained from giving completely truthful answers. Thirdly, our research was conducted on older adults with hearing loss in Tehran, thus the results may not be generalizable to other cities, age groups, countries, or cultures. Thus, it is recommended that similar studies are carried out in various cultural and social contexts and other age groups. Fourth, the present study did not investigate some potentially influential audiological factors mentioned in some other studies, such as the results of speech tests in the presence of noise (SIN) [104], real ear measurements [23], and people’s overall health condition. Future studies are thus recommended to examine these factors as well.

## 7. Conclusions

This study developed a PAM to examine the determinants of the continuous use of hearing aids among older adults in Tehran. The results of this study contribute to the existing literature by highlighting the importance of psychosocial factors in determining the continuance intention to use hearing aids among older adults. Understanding these factors can aid healthcare professionals and policymakers in developing interventions and strategies to promote the continued use of hearing aids in this population. By addressing factors such as satisfaction, self-efficacy, and social support, it may be possible to enhance the overall experience with and acceptance of hearing aids among older adults. Further research is warranted to explore additional variables that may influence continuance intentions and to validate these findings in different populations. Overall, this study provides valuable insights into improving the adoption and long-term usage of hearing aids among older adults.

## Figures and Tables

**Figure 1 healthcare-12-00487-f001:**
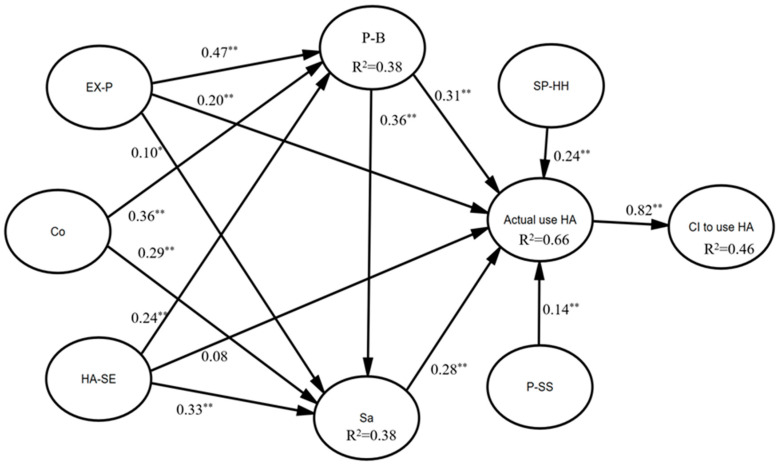
Results of path analysis. Note: *p* ˂ 0.05 = *, *p* ˂ 0.001 = **; CI = continuance intention to use hearing aid; Co = confirmation; Sa = satisfaction; P-B = perceived benefit; SP-HH = self-perceived hearing handicap; HA-SE = hearing-aid self-efficacy; EX-P = extraverted personality trait; P-SS = perceived social support; HA = hearing aid.

**Table 2 healthcare-12-00487-t002:** Participants’ demographic information (N = 300).

Variable	n (%)	Variable	n (%)
Age (mean)	71.38 (8.05)	Hearing aid usage	
60–74 years	194 (64.7%)	6–12 months	90 (30%)
74–85 years	81 (27%)	12–18 months	75 (25%)
+85 years	25 (8.3%)	˃18 months	135 (45%)
Gender		Hearing aid fitting	
Female	152 (50.7%)	Monaural	98 (32.7%)
Male	148 (49.3%)	Binaural	202 (67.3%)
Education status		Degree of hearing impairment (BEA)	
Illiterate	45 (15%)	˂26 dB HL	12 (4%)
Primary	66 (22%)	26–40 dB HL	60 (20%)
Secondary	65 (21.7%)	41–55 dB HL	97 (32.3%)
Diploma	77 (25.7%)	56–70 dB HL	104 (34.7%)
Higher education	47 (15.6%)	≥71 dB HL	27 (9%)
Marital status		Hearing aid style	
Married	211 (70.3%)		
Widow/single	71 (23.7%)	Behind the ear (BTE)	203 (67.7%)
Divorced	18 (6%)	In the ear (ITE)	97 (32.3%)
Employment status		Hearing aid type	
Employed	57 (19%)		
Retired	99 (33%)	Digital	243 (81%)
Unemployed	59 (19.7%)	Programmable	37 (12.3%)
Housewife	85 (28.3%)	Analog	20 (6.7%)

Note. BEA = better ear average (averaged over 0.5, 1, 2, and 4 kHz).

**Table 3 healthcare-12-00487-t003:** Correlation analysis of demographic variables (n = 300).

Variable	Actual Use		df	CI to Use HA	
	Mean Square	Value		Mean Square	Value
Age		r = −0.16 ***			r = 0.07 *
Gender		*t* = 1.36 *			*t* = 1.53 *
Education status	51.28	F = 12.29 ***	4	16.89	F = 2.002 *
Marital status	18.67	F = 2.96 *	2	13.87	F = 1.63 *
Employment status	20.08	F = 4.32 ***	3	7.13	F = 0.83
Degree of hearing impairment (BEA)	1.70	F = 0.35	4	1.13	F = 0.13
Hearing aid fitting		*t* = 0.24			*t* = 1.22
Style of hearing aid		*t* = −1.16			*t* = −0.08
Types of hearing aids	32.72	F = 7.09 ***	2	56.41	F = 6.87 ***
Hearing aid use	0.52	F = 0.1	2	13.04	F = 1.53

Note. *p* ˂ 0.2 = *, *p* ˂ 0.001 = ***; df = degree of freedom; CI to use HA = Continuance intention to use hearing aid; *t* = independent *t*-test; F = F-test (ANOVA); r = Pearson correlation coefficient.

**Table 4 healthcare-12-00487-t004:** Descriptive statistics of the Pearson correlation coefficients between variables.

Variable	Mean (SD)	Possible Range	CI to Use HA	Actual Use HA	Sa	P-B
1. CI to use HA	10.14 (2.91)	3–15	1			
2. Actual use HA	6.12 (2.19)	2–10	0.67 **	1		
5. SP hearing handicap	17.31 (9.13)	0–40	0.34 **	0.41 **		
6. HA self-efficacy	64.58 (16.10)	0–100	0.30 **	0.47 **		
7. P social support	45.59 (9.36)	12–60	0.40 **	0.41 **		
3. Satisfaction	65.41 (14.78)	15–105	0.50 **	0.62 **	1	
4. Perceived benefit	71.87 (14.40)	1–99	0.49 **	0.65 **	0.56 **	1
8. Extraversion Pt	7.83 (2.71)	2–14	0.37 **	0.48 **	0.39 **	0.50 **
9. Confirmation	11.67 (2.33)	3–15	0.59 **	0.60 **	0.46 **	0.47 **

Note. *p* ˂ 0.01 = **; CI to use HA = continuance intention to use hearing aid; SP = self-perceived; HA = Hearing aid; P = perceived; Pt = personality trait.

**Table 5 healthcare-12-00487-t005:** Model summary of actual use and continuance intention to use hearing aids.

Model	F	Sig	R^2^	R^2^_adj_	SE	Durbin–Watson
CI to use HA	22.91	0.001	0.60	0.57	1.91	1.93
Actual use HA	63.43	0.001	0.69	0.67	1.25	2.10

Note. SE = standard error; CI = continuance intention; HA = hearing aid.

**Table 6 healthcare-12-00487-t006:** Results of the goodness of fit test.

Indicators	Cut Point	Results	Conclusion
CMIN/DF	≤4	2.52	good fit
RMSEA	≤0.08	0.07	good fit
GFI	≥0.90	0.87	Not fit
CFI	≥0.90	0.92	good fit
TLI	≥0.90	0.90	good fit
IFI	≥0.90	0.92	good fit
RFI	≥0.60	0.84	good fit
PNFI	≥0.60	0.72	good fit
PCFI	≥0.60	0.76	good fit
AGFI	≥0.90	0.82	Not fit

Note. CMIN/DF = Chi-square statistic normalized by degrees of freedom; RMSEA = Root Mean Square Error of Approximation; GFI = goodness of fit index; CFI = comparative fit index; TLI = Tucher–Lewis index; IFI = Incremental fit index; RFI = Relative fit index; PNFI = Parsimonious normed fit index; PCFI = Parsimony comparative fit index; AGFI = Adjusted goodness of fit index.

**Table 7 healthcare-12-00487-t007:** Standardized path coefficients from the structural equation modeling analysis.

Path	Total	Direct	Indirect	SE	CR	Sig	Support
Confirmation → Perceived Benefit	0.35	0.35	0	7.35	5.34	0.001	yes
Confirmation → Satisfaction	0.41	0.29	0.12	0.22	4.23	0.002	yes
Perceived Benefit → Satisfaction	0.35	0.35	0	0.003	4.02	0.001	yes
Perceived Benefit → Actual Use	0.40	0.30	0.10	0.004	3.96	0.001	yes
Perceived Benefit → Actual Use	0.28	0.28	0	0.03	3.54	0.001	yes
HA Self-efficacy → Perceived Benefit	0.24	0.24	0	0.02	4.31	0.001	yes
HA Self-efficacy → Satisfaction	0.41	0.32	0.09	0.001	4.61	0.001	yes
HA Self-efficacy → Actual Use	0.26	0.08	0.19	0.001	1.67	0.09	no
Extraversion Pt → Actual Use	0.41	0.20	0.21	0.06	2.69	0.007	yes
Extraversion Pt → Perceived Benefit	0.46	0.46	0	5.51	5.96	0.001	yes
Extraversion Pt → Satisfaction	0.26	0.10	0.16	0.16	2.24	0.04	yes
P Social Support → Actual Use	0.14	0.14	0	0.01	3.19	0.001	yes
SP Hearing Handicap → Actual Use	0.24	0.24	0	0.01	4.97	0.001	yes
Actual Use → Continuance Intention	0.81	0.81	0	0.05	10.94	0.001	yes

Note. SP = self-perceived; HA= hearing-aid; Pt = Personality trait; P = perceived; SE = standard error; CR = critical ratio.

## Data Availability

Since this paper is extracted from a thesis, open access to the raw data is not permitted by the Medical University. If necessary, the authors can provide access to the data with the permission of the university.

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
