# Peer review of "Determinants of Continuance Intention to Use Hearing Aids among Older Adults in Tehran (Iran)"

_healthcare, 2024, doi:10.3390/healthcare12040487_

Round 1
Reviewer 1 Report
Comments and Suggestions for Authors
This manuscript consist of two different parts. One background including an extensive theoretical background. Two the actual cross sectional survey. The language and the literature review is good and comprises several theories that are of importance.
The survey. The participants were 300 hearing aid users (60 years or older) residing in Tehran. Tehran was divided into four zones, namely low, lower moderate, upper-moderate, and high in terms of socioeconomic development levels. Eventually, 75 people from each category were randomly selected (through a draw) as the research sample, 19% were employed. The data were collected using several self-evaluation tools, demographic information forms, and the last recorded audiograms of the individuals. Six different established scales were used. Table 3 presents association between different demographic factors and use of hearing aid. Table 4 presents 9 different variables some crated and their association or correlation matrix with Continuance intention to use HA, Actual use HA, Satisfaction and Perceived Benefit. Several factors were then included in a multiple regression model predicting use of hearing aid with R2= 0.69.
Table 6 summarizes the final model GOF indices and their recommended values. Table 7. Standardized path coefficients. There overall numerous abbreviations that make the text hard to follow. Figure 1 illustrate the relationships
Eventually, the results of the present study suggest that the actual use of hearing aids is a strong predictor of the continuance intention to use hearing aids. This means that the successful use of hearing aids by the elderly will increase their intention to use them continuously in the future. A result that is not entirely new but may be relevant for Iranian context. The authors discuss limitations correctly. The manuscript is very long and would benefit to be shortened considerably. There are a lot of modelling performed using questionnaire data with related factors. All this numbers make the paper difficult to follow. I can furthermore not see what new action the result might would lead to. The value of results may be limited to conditions in Teheran. My advice would be to present simplified calculations in a short paper and possibly refer to the full analysis in report available for those who want all details.
Detials Table 3 Note. * ,p˂0.05** as I can read there are no factor for this category. Table 5 difficult to understand on its own. What is included?
Comments on the Quality of English LanguageNo language problems
Author Response
We would like to extend our gratitude for the time you have taken to review this manuscript. Kindly refer to the detailed responses that have been delineated below and highlighted in red in the resubmitted manuscript.
Comments 1: {Table 7. Standardized path coefficients. There overall numerous abbreviations that make the text hard to follow}
Response 1: Thank you for pointing this out. We agree with this comment. Therefore, we have minimized the abbreviations in table 7 as much as possible so that it can be followed easily. The changes made to table 7 are highlighted in red on page 10.
Comments 2: [Note. * ,p˂0.05** as I can read there are no factor for this category..]
Response 2: Agree. Thank you for your close attention. We removed this part from table3 on page 8.
Comments 3: {Table 5 difficult to understand on its own. What is included?}
Response 3: we agree whit your opinion. Therefore, we changed the title of the table 5 and in text we gave more explanation about the content of the table for better understanding. The changes made to table 5 are highlighted in red on page 9, line 307.
5. Additional clarifications
We understand your concern about the length of the manuscript and appreciate you raising this point. In explanation, this manuscript is from a PhD thesis. Our aim was to comprehensively and accurately convey our full results for other researchers. Previous studies have usually focused on a specific aspect of factors related to hearing aid acceptance; however, our research tried to simultaneously investigate all potentially relevant factors in our models—making the article longer. We also wanted a robust study, so the background and frameworks cite various models that could aid researchers, but resulted in increased length.
Are the conclusions supported by the results?
The conclusion was edited and on page 14 it was highlighted in red.

Reviewer 2 Report
Comments and Suggestions for Authors
This study evaluated the factors influencing the continued use of hearing aids (HA) among the elderlies and found that continuance intention to use HA influenced the actual use of HA and other questionnaires also correlated with the use of HA. The results of this study are in line with the previous studies and hypothesis.
There are some small errors to be fixed before the publication.
L73 .used
L114 al.[35]
L121 model(TAM
L272 averaged71.38
L282 (2.297)
L307 (Table5)
Authors should check the all texts thoroughly during the revision process.
Author Response
We would like to extend our gratitude for the time you have taken to review this manuscript. Kindly refer to the detailed responses that have been delineated below and highlighted in red in the resubmitted manuscript.
Comments 1: {L73. used, L114 al.[35], L121 model(TAM, L272 averaged71.38, L282 (2.297), L307 (Table5) }
Response 1: Thank you for pointing out the errors. The requested corrections have been addressed in the manuscript text. L73 used: page 2, L114 al. [35],: page 3 , L121 model (TAM): page 3 , L272 averaged 71.38: page 7 , L282 (2,297); page: 7 , L307 Table5: page9

Reviewer 3 Report
Comments and Suggestions for Authors
I have had the privilege of reviewing the manuscript titled “Determinants of continuance intention to use hearing aid among older adults in Tehran (Iran)”. The study has an interesting area studying older adults and actual use of hearing aids.
Overall, I find the study to be well-conceived and worthy of publication with some revisions.
The cohort size is good with 300 hearing aid users. The study's objectives are clear. The Theoretical background is clear. The Results are a bit difficult to read both in text and tables. The manuscript use a lot of abbreviations and it becomes difficult to read the text and tables without constantly looking for explanations. Wouldn't it be better to have a list of abbreviations?
In Introduction some of references are old, and it would be recommended that use more recent references when referring, for example, to the number of people with hearing loss (ref 1, 2) in line 32.
Table 1 gives a good view of all instruments.
Table 2 has a slightly different layout, but it is probably ok for the journal. One question about Table 2 is regarding difference between Hearing aid type and “Digital” and “Programmable” - isn't this the same?
In line 301 about statistics should even mention in this test in Data analysis in line 261-270.
In line 309 more explanation about which means with the dependent and independent variables.
In Table 6 - explanations for all abbreviations are missing.
One question about hearing aid data – could researchers read the hearing aid data such as time of use from hearing aids by using data program?
From a language perspective, the manuscript is written in good English and requires no further editing what I can judge.
In conclusion, I recommend accepting this manuscript for publication after addressing the mentioned revisions.
Author Response
We would like to extend our gratitude for the time you have taken to review this manuscript. Kindly refer to the detailed responses that have been delineated below and highlighted in red in the resubmitted manuscript.
Comments 1: {The manuscript use a lot of abbreviations and it becomes difficult to read the text and tables without constantly looking for explanations. Wouldn't it be better to have a list of abbreviations?}
Response 1: Thank you for pointing this out. We agree with this comment. If you were referring to the abbreviations related to tables 4 and 7 , we have written them out in full as much as possible and highlighted them in red. Pages : 9&10
Comments 2: [In Introduction some of references are old, and it would be recommended that use more recent references when referring, for example, to the number of people with hearing loss (ref 1, 2) in line 32.]
Response 2: we agree with you that some references may be old. We have updated references 1 and 2, which are highlighted in red(page1&15). Some references pertain to foundational theories and concepts that we had to utilize.
Comments 3: { Table 2 is regarding difference between Hearing aid type and “Digital” and “Programmable” - isn't this the same?}
Response 3:In the case of programmable only the settings are changed and stored digitally. The signal is processed using analog principles only. A fully digital hearing aid utilises digital signal processing.
Comments 4: in line 301 about statistics should even mention in this test in Data analysis in line 261-270.
Response 4: we agree with you, and we have made the necessary changes that are specified in red on page 9.
Comments 5: {In line 309 more explanation about which means with the dependent and independent variables.}
Response 5: we agree with you. On page 9,we have made the requested amendments, which are highlighted in red. In this section , we have tried to to explain the variables in more understandable way.
Comments 6: In Table 6 - explanations for all abbreviations are missing.
Response 6: the full names of the abbreviations are written in table 6 and highlighted in red. Page(10)
Comments 7: {One question about hearing aid data – could researchers read the hearing aid data such as time of use from hearing aids by using data program?}
Comments 7: Yes, most digital hearing aids have data logging programs and therefore the capability to record hearing aid usage time. Since the present study also included analog hearing aids, the self-report method was used. Results from previous studies on recording hearing aid usage time have shown that there is no statistically significant difference between using the self-report method and data logged from the hearing aids.
